# Dynamics of the Tumor Immune Microenvironment during Neoadjuvant Chemotherapy of High-Grade Serous Ovarian Cancer

**DOI:** 10.3390/cancers14092308

**Published:** 2022-05-06

**Authors:** Yong Jae Lee, Ha Young Woo, Yoo-Na Kim, Junsik Park, Eun Ji Nam, Sang Wun Kim, Sunghoon Kim, Young Tae Kim, Eunhyang Park, Je-Gun Joung, Jung-Yun Lee

**Affiliations:** 1Department of Obstetrics and Gynecology, Institute of Women’s Medical Life Science, Yonsei University College of Medicine, Seoul 03722, Korea; svass@yuhs.ac (Y.J.L.); heartonbrainmd@yuhs.ac (Y.-N.K.); byjspark@yuhs.ac (J.P.); nahmej6@yuhs.ac (E.J.N.); san1@yuhs.ac (S.W.K.); shkim70@yuhs.ac (S.K.); ytkchoi@yuhs.ac (Y.T.K.); 2Department of Pathology, Kyung Hee University Hospital, Kyung Hee University College of Medicine, Seoul 02447, Korea; beliefi31@gmail.com; 3Department of Pathology, Yonsei University College of Medicine, Seoul 03722, Korea; epark54@yuhs.ac; 4Department of Biomedical Science, College of Life Science, CHA University, Seongnam 13488, Korea

**Keywords:** tumor immune microenvironment, neoadjuvant chemotherapy, tumor-infiltrating lymphocytes, high grade serous ovarian cancer, immune checkpoint inhibitors

## Abstract

**Simple Summary:**

Neoadjuvant chemotherapy (NAC) induced a dynamic change in the TIME that increased the level of immune infiltration, leading to a high number of CD8 T cells with enhanced immune activity. However, increased immune infiltration and immune activity did not present any survival benefit, probably due to concomitant immunosuppression associated with an increase in the proportion of Foxp3+ regulatory T cells. Our results could provide therapeutic strategies to improve the survival benefit from immunotherapies in an NAC setting.

**Abstract:**

The dynamic changes in the tumor immune microenvironment (TIME) triggered by neoadjuvant chemotherapy (NAC) have not been clearly defined in advanced-stage ovarian cancer. We analyzed the immunologic changes induced by NAC to correlate them with clinical outcomes. We compared the changes in the immune infiltration of high-grade serous carcinoma biopsies before and after NAC via immunohistochemistry (147 paired samples) and whole transcriptome sequencing (35 paired samples). Immunohistochemistry showed significantly increased PD-L1 levels and TIL levels after NAC. Whole transcriptome sequencing revealed that the stromal score, immune score, and cytolytic activity score significantly increased after NAC. An increased tumor-infiltrating lymphocyte (TIL) level in response to NAC was associated with shorter progression-free survival compared with decreased TIL level after NAC. In tumors with increased TIL levels after NAC, the relative fraction of CD8 T cells and regulatory T cells significantly increased with immunohistochemistry. Post-NAC tumors were enriched in gene sets associated with immune signaling pathways, such as regulatory T cell and JAK/STAT signaling pathways. NAC induced dynamic changes in the TIME that increased TIL levels, but their high abundance did not impart any survival benefit. Our data may provide therapeutic strategies to improve the survival benefit from immunotherapies in ovarian cancer.

## 1. Introduction

Ovarian cancer is one of the most lethal gynecological malignancies in women [1,2]. The encouraging results of phase 3 studies on primary chemotherapy before surgery have increased the use of neoadjuvant chemotherapy (NAC) followed by interval debulking surgery (IDS) in the treatment of advanced-stage ovarian cancer [3,4]. Approximately 30% of patients who undergo NAC have favorable progression-free survival (PFS), while the remaining 70% have high rates of relapse [5]. Thus, there is an urgent need to identify new therapeutic targets to improve the survival outcome.

Recently, immuno-oncology has been introduced to treat advanced-stage ovarian cancer, and immune checkpoint inhibitors (ICIs) have been used in combination with conventional chemo-therapeutic agents has been [6,7]. These trials are primarily based on the hypothesis that chemotherapeutic agents may actively interact with the tumor immune microenvironment (TIME) and provide a broad-acting immune stimulus against cancers [8,9,10]. However, there was no survival advantage of the combination of ICI and chemotherapy in the first-line setting for ovarian cancer patients [6,7]. Comprehensive characterization of the effects of chemotherapeutic agents on TIME and validated biomarkers are required to select ovarian cancer patients who may benefit from ICI combination with chemotherapy in frontline treatment.

To date, the effect of chemotherapy on the immunologic landscape of ovarian cancer has been studied [11,12,13,14], but it still remains unclear. Previous reports have documented that, overall, NAC increased tumor-infiltrating lymphocytes (TILs), CD3 and CD8 T cells. On the other hand, immunosuppressive cells were shown to be largely unaffected by NAC, but in one study [14] the number of CD4 T cells increased and Foxp3 remained unchanged. In breast cancer, NAC induces various changes in the TIME depending on the tumor subtype and pathological response. Patients with breast cancers harboring higher proportions of baseline and on-treatment TILs and CD8 T cells were significantly more likely to achieve complete pathologic response after NAC [15]. Understanding the effects of chemotherapy on the TIME could help in the planning of future studies and the development of biomarkers.

Here, we performed a comprehensive analysis of the TIME in pre- and post-NAC high-grade serous ovarian cancer (HGSOC) tissue from the same patient to find correlations with clinical outcomes. This study was undertaken to more fully characterize the dynamic changes of the TIME to identify an optimal immunotherapeutic strategy for ovarian cancer.

## 2. Materials and Methods

### 2.1. Study Design and Patient Samples

This study was performed on advanced-stage ovarian cancer samples with the following inclusion criteria: (1) cases with a preoperative pathological diagnosis of HGSOC (as defined by the World Health Organization [16]) via diagnostic laparoscopic biopsy; (2) cases that received at least one cycle of NAC, followed by IDS; (3) stage III/IV disease as measured by the International Federation of Gynecology and Obstetrics 2014. As a result, we collected 147 pairs of pre- and post-NAC samples. Pre-NAC samples were biopsy specimens obtained from diagnostic laparoscopy and post-NAC samples were surgically resected specimen obtained via IDS (Figure 1a). All patients were diagnosed and treated in Yonsei Cancer Center from 2010 to 2020.

This study was reviewed and approved by the Institutional Review Board (IRB) at Severance Hospital, Yonsei University Health System, Seoul, Korea (IRB number: 4-2020-0682). The IRB approved our study with a waiver of informed consent as this study presented no risk for the patients and required no interventions.

### 2.2. Treatment

All patients received NAC with a regimen consisting of platinum-based combination chemotherapy for a median of three cycles (range, 3–4 cycles). NAC was followed by IDS, and no other treatment, such as radiation or endocrine therapy, was performed before the surgery. For IDS, the intent was to achieve a complete cytoreduction with no gross residual tumor. Subsequently, additional cycles of adjuvant chemotherapy were administered to complete a total of six cycles at the discretion of the treating physician.

### 2.3. Pathological Evaluation

We retrospectively reviewed 147 pairs of pre- and post-NAC HGSOC samples. The pathological features of both groups were evaluated using hematoxylin and eosin (H&E)-stained representative sections of the formalin-fixed and paraffin-embedded (FFPE) tissue samples from each case. Two independent gynecologic pathologists (H.Y.W. and E.H.P.), who were blind to the clinical data, reviewed the slides and reached a consensus. The consensus judgements were adopted as final results.

Each case was scored between 1 and 3 using the chemotherapy response score (CRS) system, as described in a previous study [17]. In brief, CRS 1 indicates no or minimal tumor response, CRS 2 indicates an appreciable tumor response amid viable tumors, and CRS 3 indicates complete or near-complete response with no residual tumor, or scattered tumor foci with a size up to 2 mm. The level of TILs was assessed in accordance with the guidelines proposed by the International TILs Working Group in 2014 [18]. The percentage of the stromal TILs was calculated as the percentage of the area occupied by mononuclear inflammatory cells over the total intratumoral stromal area.

Immunohistochemistry (IHC) was performed on both pre-NAC HGSOC samples (whole section slides) and post-NAC samples (tissue microarrays with two different tumoral cores for each case, diameter of 3 mm). The analysis was conducted using a Ventana XT automated stainer (Ventana Medical Systems) with antibodies for PD-L1 (prediluted, clone 22C3, DAKO, Glostrup, Denmark), CD8 (prediluted, clone 4811, Novocastra Leica, Newcastle, UK), and Foxp3 (1:100, clone 236A/E7, Abcam, Cambridge, UK). The PD-L1 expression in the tumor cell membrane, and the membrane and/or cytoplasm of tumor-associated mononuclear inflammatory cells (lymphocytes and macrophages), was scored. The combined positive score (CPS) was defined as the total number of tumors and immune cells stained with PD-L1, divided by the number of all viable tumor cells and multiplied by 100. Each countable slide contained at least 100 viable tumor cells. A high PD-L1 expression was defined as having a CPS higher than 10 [19], and in cases with high PD-L1 expression, the mean percentage of TILs were calculated as 20%. As a result, the cut-off value for high TILs was set at 20%. The levels of CD8 and Foxp3 were calculated manually under high-magnification (40× objective) on 10 randomly selected fields for each sample, and the mean values of each marker were recorded.

### 2.4. Isolation of Genomic DNA and RNA

An expert pathologist (H.Y.W.) reviewed a slide with a 4-μm H&E-stained slice to ensure ≥20% of the nucleated cells in the sample were derived from the tumor. Tumor specimens were macrodissected after an H&E reference slide was checked to ensure the proportion of tumor content. Then, 10 μm-thick serial sections were cut from each paraffin block, and the macrodissected tissues were collected in 1.5 μL microcentrifuge tubes.

DNA was extracted using the RecoverAll™ Total Nucleic Acid Isolation kit (Ambion, Austin, TX, USA) in accordance with the manufacturer’s protocol. DNA was extracted from FFPE tissue via a xylene-free deparaffinization method, which consisted of the addition of 100 μL of digestion buffer and 4 μL of protease to each FFPE sample. The samples were then incubated for 16 h at 50 °C, then for 15 min at 80 °C. The sample solution of 100 μL was mixed with 120 μL of isolation additive by pipetting up and down. For each sample, the mixture (up to 700 μL) was passed through the filter cartridge during centrifugation at 10,000 rpm for 30–60 s. Wash buffers (wash 1, 700 μL, and 2/3, 500 μL) were passed through the filter cartridge, followed by an additional centrifugation step to remove residual fluid from the filter. We added 60 μL of the RNase mix to the center of each filter cartridge. This was then incubated for 30 min at room temperature (22–25 °C). Samples were washed with 700 μL of wash buffer 1, and twice with 500 μL of wash buffer 2/3, and were then centrifuged to remove residual fluid. DNA was eluted from the filter cartridge using 25 μL of the elution solution preheated to 95 °C. DNA samples were quantified using the Qubit™ dsDNA HS Assay Kits and a Qubit Fluorometer 2.0 (Invitrogen, Life Technologies, Carlsbad, CA, USA). The RNA from the cancer tissues was purified using the AllPrep RNA Mini Kit (Qiagen, Germantown, MD, USA) in accordance with the manufacturer’s protocol. The quality and quantity of the obtained RNA was checked using a NanoDrop 2000 spectrophotometer (Thermo Fisher Scientific, Wilmington, DE, USA), and analyzed with the Agilent 2100 Bioanalyzer system (Agilent Technologies, La Jolla, CA, USA). 

### 2.5. Whole Transcriptome Sequencing

A total of 35 paired samples from patients were assayed. Library construction with RNA sequencing (RNA-seq) was performed using a Truseq RNA Sample Preparation v2 Kit (Illumina). The isolated total RNA was used in a reverse transcription reaction with poly (dT) primers, using SuperScriptTM II Reverse Transcriptase (Invitrogen/Life Technologies, USA) in accordance with the manufacturer’s protocol. An RNA-seq library was prepared via cDNA amplification, end-repair, 3′-end adenylation, adapter ligation, and amplification. Library quality and quantity were measured using the Bioanalyzer and Qubit. The samples were sequenced at 101 bp using the Illumina HiSeq 2500 platform.

### 2.6. Whole Transcriptome Data Analysis

The reads from the FASTQ files were mapped against the human hg19 reference genome using the STAR software version 2.5.0a in 2-pass mode (https://github.com/alexdobin/STAR, accessed on 20 July 2021). Then, gene quantification was performed using RSEM (RNA-Seq by Expectation Maximization) (https://deweylab.github.io/RSEM/, accessed on 20 July 2021). Differentially expressed genes were identified using the DESeq R package (www.huber.embl.de/users/anders/DESeq/, accessed on 20 July 2021). A gene-set enrichment analysis (GSEA) [20] was conducted to analyze functional differences between pre- and post-NAC ovarian cancer samples. In addition, GSEA was used to estimate the enrichment scores of samples for an immune-associated gene set using the R package ‘fgsea’ (https://bioconductor.org/packages/fgsea/, accessed on 20 July 2021). Stromal and immune scores, based on whole transcriptome sequencing, were calculated using ESTIMATE (Estimation of STromal and Immune cells in MAlignant Tumor tissues using Expression data). Fractions of immune-associated cell types were calculated by quanTIseq using RNA-seq expression profiles [21]. The immune cytolytic activity (CYT) was measured by the geometric mean of GZMA and PRF1 expression values in TPM [22]. A forest plot was created using an R package to present the association between clinical factors and immune cell type fractions (https://CRAN.R-project.org/package=meta, accessed on 20 July 2021).

### 2.7. Targeted-Gene Sequencing

We performed targeted gene sequencing of 57 FFPE cancers with sufficiently high tumor cellularity (>30%). Genomic DNA was extracted using a Maxwell CSC DNA FFPE Kit (Promega, Madison, WI, USA) in accordance with the manufacturer’s instructions. The products were sequenced with a MiSeq System (Illumina, San Diego, CA, USA). Mutational and copy number analyses were performed using a TruSight Tumor 170 panel (Illumina) that covers 170 and 59 genes for mutational and copy number analyses, respectively. For mutational analysis, FASTQ files were uploaded on the Illumina BaseSpace software (Illumina) for variant interpretation. Only variants in coding and promoter regions or splice variants were retained. In addition, we retained only variants present in <1% of the population, according to ExAC and 1000 genomes, and also present in >5% of reads with a minimum read depth of 250. All retained variants were reviewed against reference websites [Catalogue of Somatic Mutations in Cancer (http://evs.gs.washington.edu/EVS/, accessed on 20 July 2021), Precision Oncology Knowledge Base (http://oncokb.org, accessed on 20 July 2021), and dbSNP (https://www.ncbi.nlm.nih.gov/, accessed on 20 July 2021)]. Only pathogenic variants were selected. In the copy number analysis, only those genes with more than a two-fold change relative to the average level were considered for amplification. We also performed total nucleic acid extraction to obtain RNA. An Archer FusionPlex Solid Tumor Kit (ArcherDx, Boulder, CO, USA), which covers 55 genes [23], was used to analyze the RNAs for fusions and splice variants. Specimens had a DNA and RNA yield of over 40 ng. Fragment sizes of DNA and RNA of at least 79 and 63 bp, respectively, were selected for targeted sequencing. Our goal in this study was to assess the feasibility and utility of using the Illumina MiSeq platform to integrate a next-generation sequencing panel into the setting of ovarian cancer clinical practice. The analyzed genes are listed in Appendix A.

### 2.8. Statistical Analysis

We used the chi-squared and the Mann–Whitney Rank Sum tests to assess the correlation between genomic alterations and clinical significance. All the tests were two-sided, and a *p*-value <0.05 was set as significant. Responses were assessed in accordance with the Response Evaluation Criteria in Solid Tumors version 1.1. We defined PFS as the time from the date of the diagnosis to disease progression or death; OS was measured from the date of the diagnosis to that of death or the last follow-up date. Survival analysis was performed using the Kaplan–Meier method with a log-rank test. All statistical analyses were performed using the R package [24].

## 3. Results

### 3.1. Study Population

The baseline characteristics of the patients are summarized in Table 1. The 147 HGSOC patients had a median age of 58 years (range, 39–81) and 75 of them (51.0%) presented stage IV disease at their first diagnosis. The median cancer antigen 125 level was 1509.0 U/mL (range, 94.2–21,994.7 U/mL) at baseline. Forty-five (30.6%) patients achieved a CRS of 3 after NAC. The median duration of follow-up was 28.2 months.

### 3.2. Changes in the Tumor Immune Microenvironment of HGSOCs before and after NAC

To evaluate the TIME-related features of HGSOCs before and after NAC, we assessed the immunohistochemical status and TIL levels in both pre- and post-NAC samples (147 pairs); a further in-depth analysis was performed with whole transcriptome sequencing (35 pairs) (Figure 1a).

According to the levels of PD-L1 expression and TIL, HGSOCs were divided into the following two subgroups: PD-L1 low/high: 81.1%/18.9%; TIL low/high: 88.3%/11.6% (Figure 1b). We compared the PD-L1 expression and TIL status in HGSOCs before and after NAC. In 25.2% of HGSOCs, the initially low PD-L1 expression increased to high; in 17.1% of HGSOCs the originally low TIL level reached a high level after NAC. Representative images of IHC staining with antibodies against PD-L1, CD8, Foxp3, and TIL density distributions in pre-NAC samples are shown in Figure 1c.

Using ESTIMATE, we found that the stromal score (*p* < 0.001), immune score (*p* < 0.001), and cytolytic activity (CYT) score (*p* = 0.021) were significantly increased after NAC (Figure 1d). The CYT score is a biomarker that can characterize the anti-tumor immune activity. With quanTIseq, we analyzed the immune cell fraction. We analyzed the correlation between immune cell fraction and maker expression (Appendix A) and the changes in the proportions of immune cell types before and after NAC (Appendix A).

### 3.3. Genetic Alterations in HGSOCs

We performed targeted gene sequencing on 57 samples of post-NAC HGSOCs. *TP53* mutation (98%), a signature alteration of the HGSOCs, was observed in most cases of our series. Besides this mutation, the most common genetic alterations were in *BRCA2* (16%) and *BRCA1* (14%), with a prevalence in frameshift or nonsense mutations, i.e., truncating mutations (Figure 2a). Twenty-five cases (44%) harbored more than one actionable alteration other than *TP53*. In 17 cases (30%), we found a mutation in at least one gene for homologous recombination deficiency (HRD), while eight had a *BRCA1* mutation, nine had a *BRCA2* mutation, two had both *BRCA1*/*2* mutations, and two had a *RAD51D* mutation. Among the 17 patients with HRD gene mutations, five patients (29%) achieved a CRS of 3, and one patient (6%) had a high PD-L1 expression with a high level of TILs.

### 3.4. Patterns of Pre-NAC Immune Microenvironment and Their Association with Outcome

We evaluated the correlation between pre-NAC TIME and survival outcome (Figure 2b). A high expression of PD-L1 pre-NAC was associated with a significant improvement in PFS (*p* = 0.0011) and OS (*p* = 0.023). A high TIL level pre-NAC was associated with a significantly improved PFS (*p* = 0.039) and a better OS trend (*p* = 0.077). We evaluated the correlation between the distribution of the densities of PD-L1 and TILs in pre-NAC samples with the CRS (Figure 2c). Tumors with high PD-L1 and TIL levels showed a good response to NAC (6/12, 50% CRS 3).

We then examined whether immune features of pre-NAC tumor samples could predict the CRS at the time of IDS. However, there were no statistically significant cell types associated with CRS (Figure 2d). In addition, we evaluated the association between the immune features of pre-NAC tumors and treatment outcome (platinum sensitivity, 12 months PFS rates). There were no immune cell types that had a statistically significant correlation with the treatment outcome, including platinum sensitivity and the PFS rate at 12 months (Appendix A).

### 3.5. Profound Impact of NAC on the Immune Microenvironment

The degree of PD-L1 expression and TIL levels significantly increased after NAC (*p* = 0.003 and *p* = 0.001, respectively), compared to before treatment. Δ TIL density (defined as: TIL [%] of post-NAC sample—TIL [%] of pre-NAC sample) and Δ PD-L1 (defined as: PD-L1 [CPS] of post-NAC sample—PD-L1 [CPS] of pre-NAC sample) were positively correlated between pre- and post-NAC tumors (ρ = 0.49, *p* = 0.003) (Figure 3a). To assess the prognostic impact of Δ PD-L1/TIL density for PFS and OS (Figure 3b), we compared the survival outcomes by classifying patients with increased PD-L1 expression and TIL density and those with decreased or no changes in tumors before and after NAC. We found no significant difference in the survival outcome with an increase in PD-L1 expression. However, the PFS (*p* = 0.01) and OS (*p* = 0.027) were significantly lower in patients with an increased TIL density. We then assessed the correlation between changes in CD8 and Foxp3 and the TIL level with an immunostaining analysis in matched samples before and after NAC (Figure 3c). Both CD8+ T cells and Foxp3+ T cells were positively correlated with increasing TILs (CD8+: ρ = 0.51 and *p* = 0.002; Foxp3+: ρ = 0.49, *p* = 0.003) (Figure 3d).

### 3.6. Differentially Expressed Gene Analysis and Gene Set Enrichment Analysis in Pre- and Post- NAC Samples

A total of 60 genes were identified by differentially expressed gene analyses in pre- and post-NAC tumors, with pre-defined criteria (>two-fold change and adjusted *p* < 0.01) (Figure 4a). Among them, seven genes (*FHL1*, *SELL*, *VIM*, *IGFBP7*, *TNFAIP3*, *LCP2*, and *SLC7AB*) were associated with the cancer-immunity cycle. In gene set enrichment analysis, several immune-related gene sets, including JAK/STAT signaling and CD28 co-stimulation, were enriched in post-NAC tumors (Figure 4b and Appendix A). In addition, gene set enrichment analysis showed that the regulatory T cell-related gene set was enriched in post-NAC tumors (Figure 4c).

## 4. Discussion

In this study, we performed a comprehensive analysis of TIME changes of advanced-stage ovarian cancer in response to NAC. We utilized 147 matched pairs of HGSOC samples, and evaluated various aspects of the TIME of ovarian cancers using histological, immune-histochemical, and in-depth DNA/RNA sequencing analyses. We observed that the degree of baseline immune infiltrate before NAC, in particular that of TIL and PD-L1, was associated with improved survival outcomes. NAC induced a dynamic change in the TIME that increased the extent of immune infiltration in the HGSOCs, leading to a high number of CD8 T cells with enhanced immune activity. However, increased TIL infiltration and immune activity did not improve the survival outcome. We found that the change in TIME during NAC not only resulted in immune stimulation, but also in immune suppression (associated with the increase in the proportion of Foxp3+ Tregs). Therefore, our data may imply that the enhanced immunosuppressive activity of Tregs induced by NAC inhibits anti-tumor immunity and imparts a negative impact for prognosis. From a clinical point of view, these results suggested that agents targeting Tregs in adjunction with conventional agents and ICIs may be useful to improve the survival outcome in advanced-stage ovarian cancer patients.

In recent decades, with growing interest in tumor immunology and immunotherapy, studies on TIME in ovarian cancer have been widely conducted. Ovarian cancer has traditionally been considered as having “cold tumors”, characterized by a low tumor mutation burden and scarce immunogenicity [25]. Previous studies reported that high levels of TILs and CD8 T cells infiltration significantly improved ovarian cancer patients’ survival outcomes [26,27,28,29,30,31,32,33]. Likewise, our series indicated that a high level of pre-NAC TIL was associated with improved survival and higher rates of CRS 3.

The TIME can be altered by various external factors, including chemotherapeutic agents, by activating antitumor immune responses or by depleting immunosuppressive cells [34]. Many studies about the impact of NAC on TIME in ovarian cancer have been published [11,12,13,35,36,37], and have generally demonstrated that the levels of CD3/CD8 T cells and CD68 tumor-associated macrophages increased after NAC, while Foxp3 regulatory T cells decreased after NAC. In contrast, there are conflicting results regarding the association between the changes in TILs after NAC and survival in ovarian cancer. Charlotte et al. [12] analyzed the matched pre- and post-NAC samples from 26 HGSOC patients via IHC. NAC was associated with increased densities of CD3, CD8, CD8 TIA-1, PD-1 and CD20 TIL. The abundance of immunosuppressive cell types, including Foxp3, IDO-1, and CD68 PD-L1 macrophages, was unchanged. However, increased TIL levels showed limited prognostic significance. Kim et al. [36] reported the prognostic value of PD-L1, CD8+ or Foxp3 + TILs, as well as the expression of immune checkpoint markers on immune cells in residual tumors after NAC. High expression levels of immune factors did not significantly impact the survival outcome. These results suggest that an increased level of TILs after NAC may not result in any survival benefit.

Interestingly, in our series, HGSOCs with increased TILs after NAC were associated with a poor survival outcome. These results are inconsistent with recent studies in ovarian cancer for the following reasons. First, Leary et al. [35] and Mesnage et al. [11] analyzed not only pre- and post-NAC matched samples, but also relapse samples. Pre-NAC and post-chemotherapy and/or at relapse samples were analyzed. In addition, their results include various subtypes of epithelial ovarian cancer, which may affect the results. Second, Charlotte et al. [12] and Kim et al. [36] analyzed the pre- and post-NAC matched samples, but the sample size was too small in each of the studies. Charlotte et al. included only 26 HGSOC patients and Kim et al. included 131, but among them only 76 (58%) had matched samples. In our study, we comprehensively evaluated the dynamics of the tumor microenvironment of HGSOCs during NAC, by pathological (in 147 paired samples) and transcriptional (in 35 paired samples) analyses.

As NAC increased not only TIL density but also the fraction of Tregs, T cell priming and the reinvigoration of ICI-exhausted T cells may improve the response to NAC and survival outcome. Tregs play a significant immunosuppressive role, inhibiting antitumor immunity and suppressing the function of effector T cells. Therefore, the activity of Tregs in the TIME may be a potential therapeutic target, and combining Tregs-targeted therapies with other treatments, including ICI and chemotherapy, can provide a synergistic antitumor effect, overcoming the immunosuppression in TIME. However, knowledge about the best timing and sequencing of the combination of ICIs and chemotherapy as a first-line treatment is still limited. Our study showed that NAC modulates the TIME not only by increasing the abundance of TILs but also by increasing that of Tregs. These results suggest that the immunomodulatory effects of NAC are variable and likely reflect the heterogeneity of ovarian cancer. JAVELIN 100 and IMAGYn050 trials showed no survival benefit in adding ICIs to a maintenance treatment after first-line therapy or adding anti-PD-1 mono-therapy to conventional frontline chemotherapy in ovarian cancer. These results suggest that re-activating the anti-tumor immune response may require targeting more than one step in the immunity cycle in ovarian cancer. Therefore, the combination of anti-PD-1/PD-L1- and Tregs-targeted agents together with NAC may enhance the anti-tumor immune response in ovarian cancer patients. Clinical trials on the combination of ICI (anti-PD-L1 and anti-CTLA-4) treatment with platinum-based chemotherapy in NAC are currently ongoing—such trials may be crucial in establishing their synergistic effect and survival benefit [38]. The results of this clinical trial may validate our study results.

Our study, however, has two limitations. First, as the anatomical sites of pre-NAC and post-NAC samples are different in many cases, it is difficult to conclude that the difference of the TIME is only influenced by the NAC. The pre-NAC samples were obtained from diagnostic laparoscopy; whereas the post-NAC samples were surgically resected during IDS, and the FFPE blocks with the most abundant residual tumors regardless of their site were selected. Second, most of the pre-NAC samples are very small in size, so the issue of tumor heterogeneity cannot be excluded.

## 5. Conclusions

In conclusion, this study elaborated on the dynamic changes in the TIME caused by NAC and provides prognostic information according to these changes in advanced-stage ovarian cancer patients. We observed that high pre-NAC TIL density and PD-L1 expression are associated with better survival outcomes. NAC increased TIL density and CD8 T cells in ovarian cancer, but increased TIL had a significant negative impact on prognosis. Of note, we observed that NAC induced the infiltration of Foxp3 T cells, implying the need for further research on agents targeting immune-suppressive mechanism. Our results could provide rationale for clinical trials evaluating ICIs combined with platinum-based chemotherapy for the treatment of advanced-stage ovarian cancer in an NAC setting.

## Figures and Tables

**Figure 1 cancers-14-02308-f001:**
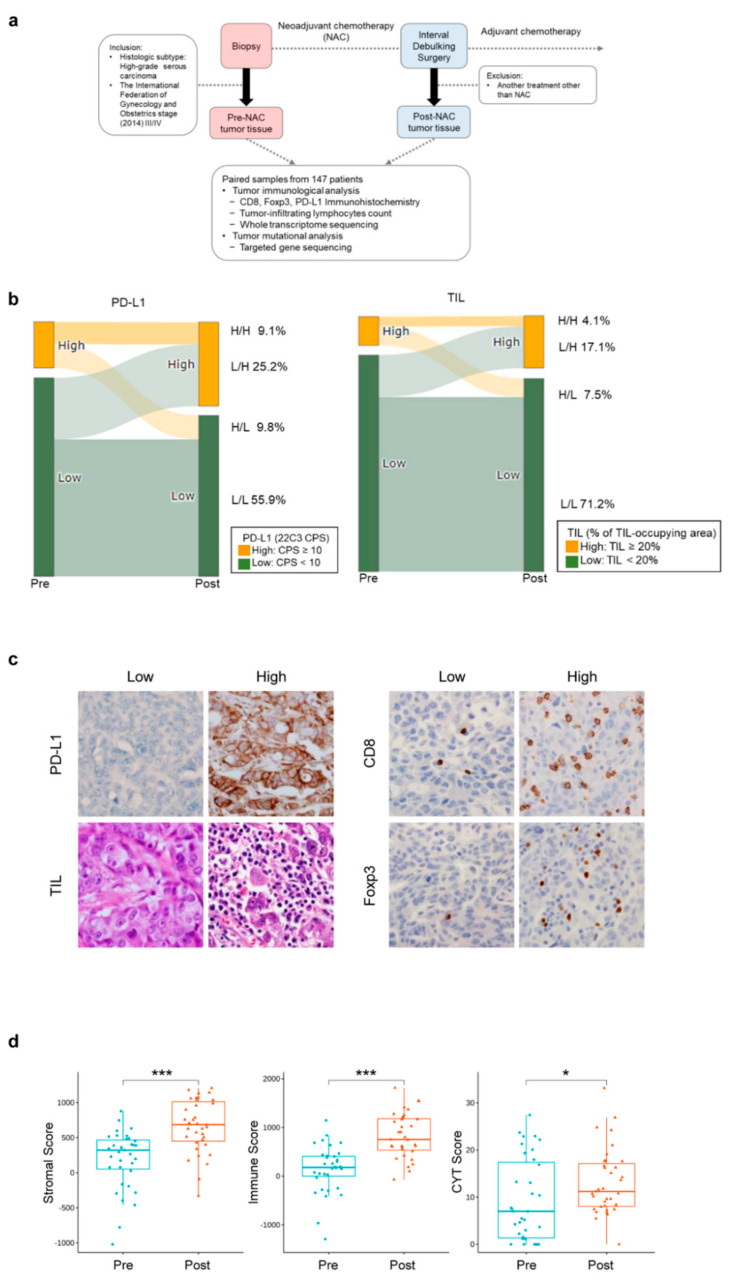
Changes in the tumor immune microenvironment in HGSOC during NAC. (**a**) Study schema. (**b**) Sankey plot of immune stage change from pre-NAC to post-NAC via immunohistochemical staining. (**c**) Representative images of low and high immunohistochemical expression for PD-L1, CD8, Foxp3, and TIL density distribution in ovarian cancer (original magnification; 40× objective). (**d**) The immune and stromal scores, and immune cytolytic activity were compared between pre- and post-NAC tumor samples. Asterisks indicate statistical significance based on multiple regression adjusting for subtype as a covariate (Stromal Score, *p* = 3.0 × 10^−6^; Immune Score, *p* = 5.3 × 10^−6^; CYT Score, *p* = 0.021, Wilcoxon test). HGSOC, high grade serous ovarian cancer; NAC, neoadjuvant chemotherapy; TIL, Tumor-infiltrating lymphocyte. * *p* < 0.05, *** *p* < 0.001.

**Figure 2 cancers-14-02308-f002:**
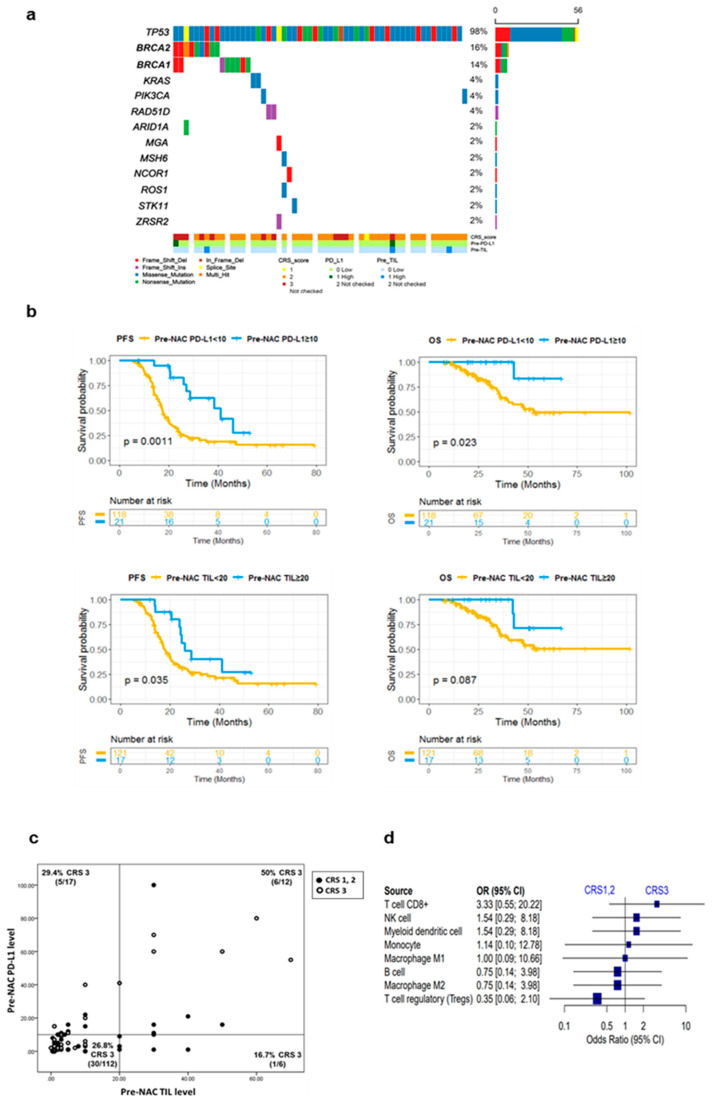
Patterns in the pre-NAC tumor immune microenvironment and their association with treatment outcome. (**a**) Landscape of mutation profiles (Oncoplot). (**b**) Prognostic values of pre-NAC PD-L1 and TIL densities as indicated by the Kaplan–Meier curves of PFS and OS. (**c**) Distribution of tumors by pre-NAC PD-L1 and TIL densities, with the CRS. Lines indicate the baseline for each level, and proportions of CRS 3 for each quartile above/below the baseline are indicated in each corner. (**d**) Forest plot showing the association between immune cell fractions and the CRS. The *x*-axis shows the log odds ratio of% cell fractions in CRS 1 and 2 versus CRS 3. TMB, tumor mutation burden; NAC, neoadjuvant chemotherapy; TIL, tumor-infiltrating lymphocyte; PFS, progression-free survival; OS, overall survival; CRS, chemotherapy response score.

**Figure 3 cancers-14-02308-f003:**
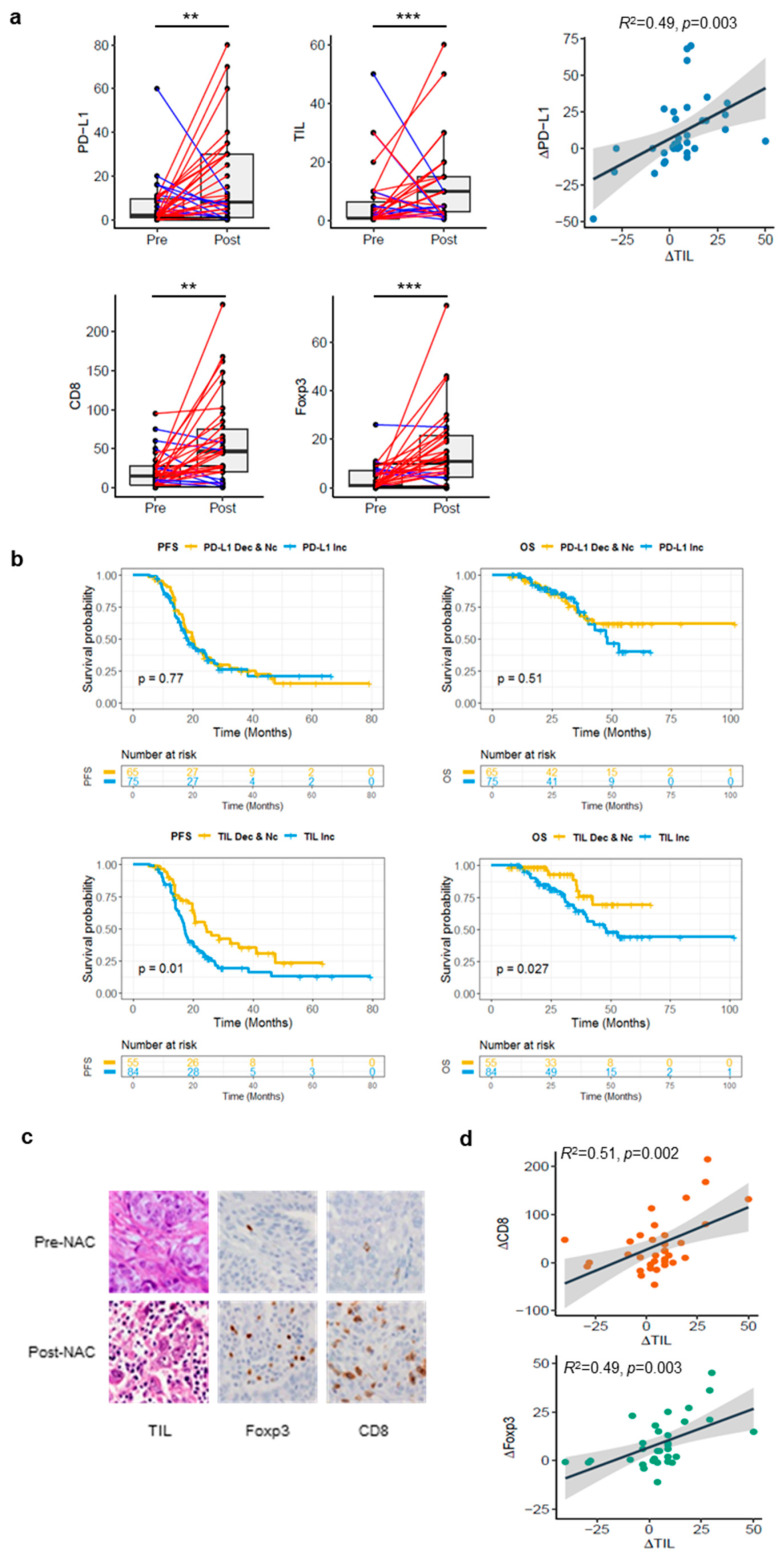
Comparison of changes in the tumor immune microenvironment between matched samples. (**a**) Comparison of PD-L1 and TIL densities in pre- and post-NAC samples based on immunohistochemical staining. Scatterplots with linear regression line and shaded 95% confidence region between PD-L1 and TIL changes. (**b**) Prognostic value of PD-L1 and TIL density change as indicated by the Kaplan–Meier curves of PFS and OS. (**c**) Representative images of pre- and post-NAC TILs, Foxp3 and CD8 immunostaining (original magnification; 40× objective). (**d**) Scatterplots with linear regression line and shaded 95% confidence region between TIL level and CD8, Foxp3 staining intensity. ** *p* < 0.01, *** *p* < 0.001.

**Figure 4 cancers-14-02308-f004:**
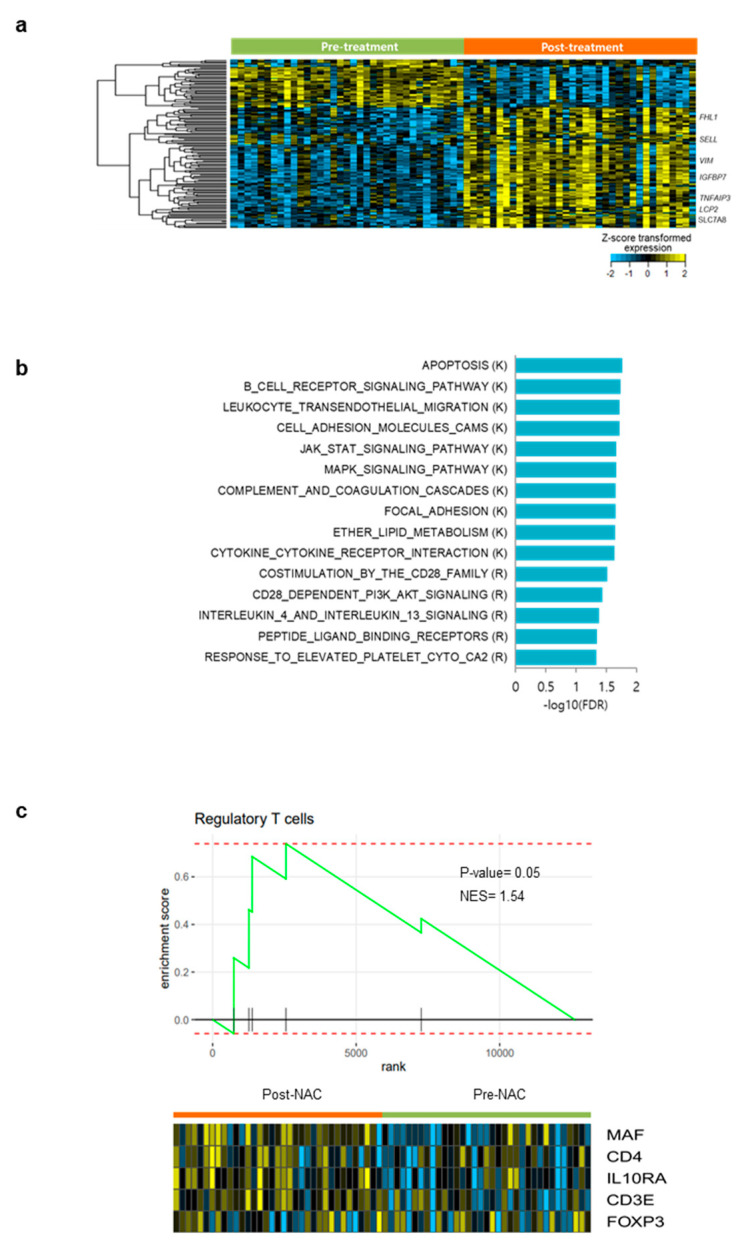
Differentially expressed genes and gene set enrichment analysis in pre- and post-NAC samples. (**a**) Heatmap comparison of 6 significant genes (>two-fold change and adjusted *p* < 0.01) between pre- and post-NAC samples. (**b**) Enriched differentially expressed gene sets between pre- and post-NAC. (**c**) Gene set enrichment analysis of differentially expressed genes of regulatory T cells in pre- and post-NAC samples. NAC, neoadjuvant chemotherapy.

**Table 1 cancers-14-02308-t001:** The baseline characteristics of the patients (*n* = 147).

Variables	Number (%) or Median (Range)
Age (years)	58 (39–81)
FIGO stage	
III	72 (49.0)
IV	75 (51.0)
ASA score before NAC	
1	8 (5.4)
2	80 (54.4)
3	58 (39.5)
Not applicable	1 (0.7)
Germline BRCA status	
Wild	97 (66.0)
BRCA1/2 mutation	36 (24.5)
Not applicable	14 (9.5)
Serum CA-125 level (U/mL)	1509.0 (94.2–21,994.7)
Cycles of NAC	3 (3–4)
Chemotherapy response score	
1	4 (2.7)
2	98 (66.7)
3	45 (30.6)
Residual disease after IDS	
No gross tumor	73 (49.7)
≤0.5 cm	53 (36.0)
>0.5 cm and ≤1.0 cm	21 (14.3)

## Data Availability

The raw data supporting the conclusions of this article will be made available by the authors without undue reservation.

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
