# Peer review of "Dynamics of the Tumor Immune Microenvironment during Neoadjuvant Chemotherapy of High-Grade Serous Ovarian Cancer"

_cancers, 2022, doi:10.3390/cancers14092308_

Round 1
Reviewer 1 Report
The authors analysed high-grade serous ovarian cancer tumor microenvironment by immunohistochemistry and bulk RNA-sequencing from 147, and 35 (?) pairs of pre-post neoadjuvant chemotherapy (NACT) specimens, respectively. The patient cohort is carefully selected with only late stage HGSOC, and the pathological assessment is well described and solid. However, the RNA-sequencing analysis, especially the cell type deconvolution results using CIBERSORT seem unrealistic and should be further validated by marker gene comparison and possibly replaced with a more accurate method or simply discarded.
Major comments:
- The cibersort analysis results (Fig S1, used further in Fig 2D, Fig S2, Fig 3C) do not sound even remotely plausible, showing memory B cells, plasma B cells and even mast cells as the most prevalent immune cell types. The authors should further show how expression levels of specific markers for e.g. CD8+ T cells (CD8A), plasma cells (MZB1) and mast cells (TPSAB1) correlate with CIBERSORT estimated cell type abundance within their data set. For CD8 T cells and Tregs they should also compare CIBERSORT estimates to matched IHC estimates on CD8 and FOXP3 levels. If the CIBERSORT results will match poorly to marker expression and/or IHC estimates, the authors should either test another deconvolution method that they can show to produce marker/IHC matching estimates, or simply compare cell type marker levels instead of flawed cell type estimates. This is a critical issue with the paper, and if not corrected, the paper should not be accepted for publication.
- The authors do not justify the thresholds used for TIL or PD-L1 low vs high patients. As the major claims of the paper are based on this, the thresholds should be clearly justified.
- There are conflicting claims of the number of RNA-seq analysed samples between Results and Discussion. Discussion suggests that 147 pairs were analysed by RNA-seq, while in the Results section the authors mention that RNA-seq analysis was performed for 35 pairs:
Discussion: “In our study, we evaluated the changes in immune infiltrate before and after NAC using 147 matched HGSOC samples by IHC and whole transcriptome sequencing.”
Results: “To evaluate the TIME-related features of HGSOCs before and after NAC, we assessed 222 the immunohistochemical status and TIL levels in both pre- and post-NAC samples (147 pairs); a further in-depth analysis was performed by whole transcriptome sequencing (35 pairs) (Figure 1A).”
The authors should correct the RNA-seq sample numbers in Discussion/Results.
Minor comments:
- The authors should remove TGFB1 from the markers they show in Fig 4C for Tregs as TGFB1 is very widely expressed across immune and stromal cell types.
- The authors could analyse further the unexpected poor survival associated with increased TILs by using a Cox regression model to assess if TILs associate with poor survival also independent of Foxp3.
Author Response
Reviewer 1
The authors analysed high-grade serous ovarian cancer tumor microenvironment by immunohistochemistry and bulk RNA-sequencing from 147, and 35 (?) pairs of pre-post neoadjuvant chemotherapy (NACT) specimens, respectively. The patient cohort is carefully selected with only late stage HGSOC, and the pathological assessment is well described and solid. However, the RNA-sequencing analysis, especially the cell type deconvolution results using CIBERSORT seem unrealistic and should be further validated by marker gene comparison and possibly replaced with a more accurate method or simply discarded.
: We appreciate this positive statement regarding the overall evaluation. Following the constructive comments and suggestions, we have revised the entire manuscript and provide the point-by-point responses as follows.
Major comments:
- The cibersort analysis results (Fig S1, used further in Fig 2D, Fig S2, Fig 3C) do not sound even remotely plausible, showing memory B cells, plasma B cells and even mast cells as the most prevalent immune cell types. The authors should further show how expression levels of specific markers for e.g. CD8+ T cells (CD8A), plasma cells (MZB1) and mast cells (TPSAB1) correlate with CIBERSORT estimated cell type abundance within their data set. For CD8 T cells and Tregs they should also compare CIBERSORT estimates to matched IHC estimates on CD8 and FOXP3 levels. If the CIBERSORT results will match poorly to marker expression and/or IHC estimates, the authors should either test another deconvolution method that they can show to produce marker/IHC matching estimates, or simply compare cell type marker levels instead of flawed cell type estimates. This is a critical issue with the paper, and if not corrected, the paper should not be accepted for publication.
Answer: As the referee commended, we have tested if there are good correlations between the cell type abundances and the expression of corresponding markers. The deconvolution by CIBERSORT did not show good correlations between both, especially between the fraction of regulatory T cell and FOXP3 expression. Thus, we have applied more reliable deconvolution method, quanTIseq (Finotello et al, Genome Medicine, 2019). Using quanTIseq, T-cell CD8 fraction showed a strong correlation with CD8A expression (correlation coefficient = 0.96). Also Treg fraction had a good correlation with FOXP3 expression (correlation coefficient = 0.61). We have updated figures from estimates of cell fractions by quanTIseq.
We revised the following sentences in the methods (Line 174 in revised manuscript)
“Fractions of immune-associated cell types were calculated by CIBERSORT using RNA-seq expression profiles [23].” (original version)
↓
“Fractions of immune-associated cell types were calculated by quanTIseq using RNA-seq expression profiles [23]. (revised version)
Reference
[23] Newman, A.M.; Liu, C.L.; Green, M.R.; Gentles, A.J.; Feng, W.; Xu, Y.; Hoang, C.D.; Diehn, M.; Alizadeh, A.A. Robust enumeration of cell subsets from tissue expression profiles. Nat Methods 2015, 12, 453-457. (original version)
↓
[23] Finotello F, et al, Ericsson-Gonzalez P, Charoentong P, Balko J, de Miranda NFDCC, Trajanoski Z. Molecular and pharmacological modulators of the tumor immune contexture revealed by deconvolution of RNA-seq data. Genome Medicine, 2019. 11(1):34. (revised version)
We revised the following sentences in the results (Line 235 in revised manuscript)
“With CIBERSORT, we analyzed the changes in the proportions of immune cell types before and after NAC (Supplemental Figure S1).” (original version)
↓
“With quanTIseq, we analyzed the immune cell fraction. We analyzed the correlation between immune cell fraction and maker expression (Supplemental Figure S1A) and the changes in the proportions of immune cell types before and after NAC (Supplemental Figure S1B).” (revised version)
We removed the following sentences in the results (Line 295 in revised manuscript) and Figure 3C
“We then assessed the correlation between changes in PD-L1 and TILs and immune cell fraction by transcriptome sequencing in matched samples before and after NAC (Figure 3C, D). Changes in the fraction of regulatory T cells (Tregs) (ρ = 0.311, p = 0.068), and CD8 T cells (ρ = 0.038, p = 0.826) showed positive correlation with increased TIL density, while dendritic cells (ρ = −0.316, p = 0.064) and M1 macrophages cells (ρ = −0.081, p = 0.641) showed negative correlation with the increased TIL density. Likewise, in the immunostaining analysis, both CD8+ T cells and Foxp3+ T cells were positively correlated with increasing TILs (CD8+: ρ = 0.51 and p = 0.002; Foxp3+: ρ = 0.49, p = 0.003) (Figure 3E).” (original version)
↓
“We then assessed the correlation between changes in CD8 and Foxp3 and TIL level by immunostaining analysis in matched samples before and after NAC (Figure 3C). Both CD8+ T cells and Foxp3+ T cells were positively correlated with increasing TILs (CD8+: ρ = 0.51 and p = 0.002; Foxp3+: ρ = 0.49, p = 0.003) (Figure 3D).” (revised version)
In CIBERSORT result, the faction of B cells was relatively higher than other cell types. B cell abundance was also shown in the result by quanTIseq. However, it is not considerable compared to total of uncharacterized or undetectable cells. When taking a look at the results of single cell study of ovarian cancers, it suggests that ratios of plasma B cells and B cells could be higher or similar to other immune cell types, except for non-immune cell types (such as epithelial cells, endotherial cells and fibroblasts) (Olalekan et al, Cell Reports, 2021).
Reference
Olalekan S, et al, Characterizing the tumor microenvironment of metastatic ovarian cancer by single-cell transcriptomics, Cell Reports, 2021. 35(8):109165.
- The authors do not justify the thresholds used for TIL or PD-L1 low vs high patients. As the major claims of the paper are based on this, the thresholds should be clearly justified.
Answer: We appreciate your kind comment. Although it is well known that the level of TILs and PD-L1 expression impact the prognosis, a consistent cut-off value for the two variables has not been established. In a large-scaled phase II study of pembrolizumab (NCT02674061), high PD-L1 expression (defined as CPS ≥ 10) showed a significant correlation with prognosis. In accordance with it, we applied CPS 10 as a cut-off value for high PD-L1 expression, and in cases with high PD-L1 expression, the mean percentage of TILs were calculated as 20%. As a result, we set cut-off values of TILs and PD-L1 expression as 20% and CPS 10, respectively.
We revised the following sentences in the methods (Line 108 in revised manuscript)
“The percentage of the stromal TILs was calculated as the percentage of the area occupied by mononuclear inflammatory cells over the total intratumoral stromal area, and high levels of stromal TILs were defined with an infiltration ≥ 20% [20].” (original version)
↓
“The percentage of the stromal TILs was calculated as the percentage of the area occupied by mononuclear inflammatory cells over the total intratumoral stromal area” (revised version)
Line 121 in revised manuscript
“The high PD-L1 expression CPS was found to be higher than 10 [21]. The levels of CD8 and Foxp3 were calculated manually under high-magnification (40× objective) on 10 ran-domly selected fields for each sample, and the mean values of each marker were recorded.” (original version)
↓
“The high PD-L1 expression was defined as CPS higher than 10 [20], and in cases with high PD-L1 expression, the mean percentage of TILs were calculated as 20%. As a result, the cut-off value for high TILs was set at 20%. The levels of CD8 and Foxp3 were calculated manually under high-magnification (40× objective) on 10 randomly selected fields for each sample, and the mean values of each marker were recorded.” (revised version)
- There are conflicting claims of the number of RNA-seq analysed samples between Results and Discussion. Discussion suggests that 147 pairs were analysed by RNA-seq, while in the Results section the authors mention that RNA-seq analysis was performed for 35 pairs:
Discussion: “In our study, we evaluated the changes in immune infiltrate before and after NAC using 147 matched HGSOC samples by IHC and whole transcriptome sequencing.”
Results: “To evaluate the TIME-related features of HGSOCs before and after NAC, we assessed the immunohistochemical status and TIL levels in both pre- and post-NAC samples (147 pairs); a further in-depth analysis was performed by whole transcriptome sequencing (35 pairs) (Figure 1A).”
The authors should correct the RNA-seq sample numbers in Discussion/Results.
Answer: Thank you for pointing out the need of clarification. We performed the whole transcriptome sequencing for 35 pairs to evaluate the TIME-related features of HGSOCs before and after NAC. Following the comment, we revised the manuscript in discussion section.
We added the following sentences in the methods (Line 370 in revised manuscript)
“In our study, we evaluated the changes in immune infiltrate before and after NAC using 147 matched HGSOC samples by IHC and whole transcriptome sequencing.” (original version)
↓
“In our study, we comprehensively evaluated the dynamics of the tumor microenvironment of HGSOCs during NAC, by pathological (in 147 paired samples) and transcriptional (in 35 paired samples) analyses.” (revised version)
Minor comments:
- The authors should remove TGFB1 from the markers they show in Fig 4C for Tregs as TGFB1 is very widely expressed across immune and stromal cell types.
Answer: Thank you for the thoughtful comment. As the reviewer commended, we have removed TGFB1 from the markers and have added another well-known Treg marker, IL10RA.
- The authors could analyse further the unexpected poor survival associated with increased TILs by using a Cox regression model to assess if TILs associate with poor survival also independent of Foxp3.
Answer: Thank you for your kind comment. In our study, increased TILs and immune activity did not improve the survival outcome. Our data may imply that enhanced immunosuppressive activity of Tregs induced by NAC inhibits the anti-tumor immunity and influences the negative impact on survival outcomes. As the reviewer commended, we performed the Cox regression in 35 matched samples. In the cox regression model, increased TIL infiltration also showed a negative impact on prognosis.
Multivariate analyses for progression-free and overall survival using a Cox proportional hazards model in 35 matched samples
|
Variables |
PFS |
|
OS |
|
|
|
HR (95% CI) |
p |
HR (95% CI) |
p |
|
Age, years |
|
|
|
|
|
≤58 |
1 |
|
1 |
|
|
>58 |
0.40 (0.14-1.13) |
0.082 |
0.18 (0.02-1.71) |
0.135 |
|
FIGO stage |
|
|
|
|
|
III |
1 |
|
1 |
|
|
IV |
0.17 (0.02-1.44) |
0.104 |
0.13 (0.01-1.47) |
0.10 |
|
CRS |
|
|
|
|
|
1-2 |
1 |
|
1 |
|
|
3 |
1.08 (0.36-3.27) |
0.891 |
0.37 (0.15-0.93) |
0.035 |
|
Residual disease |
|
|
|
|
|
No |
1 |
|
1 |
|
|
Any residual |
1.01 (0.40-2.56) |
0.976 |
7.31 (0.47-114.32) |
0.156 |
|
TILs changes |
|
|
|
|
|
Decreased or no change |
1 |
|
1 |
|
|
Increased |
3.99 (1.00-15.90) |
0.050 |
12.71 (1.12-143.60) |
0.040 |
|
PD-L1 changes |
|
|
|
|
|
Decreased or no change |
1 |
|
1 |
|
|
Increased |
0.35 (0.11-1.07) |
0.065 |
6.15 (0.62-60.65) |
0.120 |
|
CD8 changes |
|
|
|
|
|
Decreased or no change |
1 |
|
1 |
|
|
Increased |
0.50 (0.12-2.05) |
0.333 |
4.39 (0.66-290.54) |
0.490 |
|
Foxp3 changes |
|
|
|
|
|
Decreased or no change |
1 |
|
1 |
|
|
Increased |
1.95 (0.55-6.90) |
0.298 |
0.99 (0.07-13.71) |
0.993 |

Reviewer 2 Report
Dynamics of the tumor immune microenvironment during neoadjuvant chemotherapy of high-grade serous ovarian cancer by Yong Jae Lee an collegues,
Is an interesting report regarding the immunological checkpoint in ovarian cancer. the article is well written, the methodology is on point, the figures represent the results obtained, and the conclusion is well based on the results
Therfor I give my favorable opinion to consider this report for publication.
Author Response
: We would like to express our gratitude to the editor and the reviewers for taking the time to carefully read and consider the report on our present study. We appreciate the opportunity to revise the manuscript by addressing the reviewers’ comments based on their constructive guidance to help you and the reviewers in your final decision.

Reviewer 3 Report
This is the first study analyzing the dynamic changes in the tumor immune microenvironment (TIME) triggered by neo-adjuvant chemotherapy (NAC) in advanced-stage ovarian cancer. Research design is clear and appropriate to aims of the study. Statistical power is acceptable. The observation that increased tumor-infiltrating lymphocyte (TIL) level in response to NAC is associated with shorter progression-free survival compared with decreased
TIL level after NAC is interesting. may provide therapeutic
strategies to improve survival benefit from immunotherapies in ovarian cancer. Such approach could delineate therapeutic strategies to improve survival benefit from immunotherapies in NAC setting.
Author Response

(The authors gave the same response as above.)

Round 2
Reviewer 1 Report
First, I want to thank the authors for taking a very careful advice of the review comments. Most importantly, they performed extensive re-analysis of bulk RNA-seq data, wherein they applied a new deconvolution method after failed sanity check of CIBERSORT results. This and other changes in both analysis and text have made the results much more robust and clearly improved the manuscript which is now almost ready for publication.
I have only a few minor comments left:
- Clearly define the sample numbers for IHC & RNA-seq also in the abstract:
“We compared the changes in immune infiltrate before and after NAC in 147 high-grade serous carcinoma matched biopsies via immunohistochemistry and whole transcriptome sequencing”
Correct to =>
“We compared the changes in immune infiltration of high-grade serous carcinoma biopsies before and after NAC via immunohistochemistry (147 paired samples) and whole transcriptome sequencing (35 paired samples).
2. Briefly define/explain the term “CYT score”, p. 6, line 235.
Author Response
Reviewer 1
First, I want to thank the authors for taking a very careful advice of the review comments. Most importantly, they performed extensive re-analysis of bulk RNA-seq data, wherein they applied a new deconvolution method after failed sanity check of CIBERSORT results. This and other changes in both analysis and text have made the results much more robust and clearly improved the manuscript which is now almost ready for publication. I have only a few minor comments left:
: We appreciate this positive statement regarding the overall evaluation. Following the constructive comments and suggestions, we have revised the entire manuscript and provide the point-by-point responses as follows.
- Clearly define the sample numbers for IHC & RNA-seq also in the abstract:
Answer: As the referee commended, we have added the sample number for IHC & RNA-seq in the abstract.
“We compared the changes in immune infiltrate before and after NAC in 147 high-grade serous carcinoma matched biopsies via immunohistochemistry and whole transcriptome sequencing” (original version)
↓
“We compared the changes in immune infiltration of high-grade serous carcinoma biopsies before and after NAC via immunohistochemistry (147 paired samples) and whole transcriptome sequencing (35 paired samples). (revised version)
- Briefly define/explain the term “CYT score”, p. 6, line 235.
Answer: Thank you for pointing out the need of clarification. As the referee commended,
We have revised the following sentences in the manuscript (Line 234 in revised manuscript)
“Using ESTIMATE, we found the stromal score (p < 0.001), immune score (p < 0.001), and CYT score (p = 0.021) were significantly increased after NAC (Figure 1D).” (original version)
↓
“Using ESTIMATE, we found the stromal score (p < 0.001), immune score (p < 0.001), and cytolytic activity (CYT) score (p = 0.021) were significantly increased after NAC (Figure 1D). CYT score is a biomarker that can characterize the anti-tumor immune activity.” (revised version)
